# Synergistic Strategies for Gastrointestinal Cancer Care: Unveiling the Benefits of Immunonutrition and Microbiota Modulation

**DOI:** 10.3390/nu15204408

**Published:** 2023-10-17

**Authors:** Serena Martinelli, Ingrid Lamminpää, Eda Nur Dübüş, Dilara Sarıkaya, Elena Niccolai

**Affiliations:** 1Department of Experimental and Clinical Medicine, University of Florence, 50134 Firenze, Italy; serena.martinelli@unifi.it (S.M.); ingrid.lamminpaa@gmail.com (I.L.); 2Department of Nutrition and Dietetics, Gazi University, 06560 Ankara, Turkey; edanurdubus@gmail.com (E.N.D.); dilarasrky2@gmail.com (D.S.)

**Keywords:** gastrointestinal cancers, nutritional therapy, microbiota, immunonutrition, probiotics, immunonutrients

## Abstract

Gastrointestinal (GI) cancers are a group of highly prevalent malignant tumors affecting the gastrointestinal tract. Globally, one in four cancer cases and one in three cancer deaths are estimated to be GI cancers. They can alter digestive and absorption functions, leading to severe malnutrition which may worsen the prognosis of the patients. Therefore, nutritional intervention and monitoring play a fundamental role in managing metabolic alterations and cancer symptoms, as well as minimizing side effects and increasing the effectiveness of chemotherapy. In this scenario, the use of immunonutrients that are able to modulate the immune system and the modification/regulation of the gut microbiota composition have gained attention as a possible strategy to improve the conditions of these patients. The complex interaction between nutrients and microbiota might contribute to maintaining the homeostasis of each individual’s immune system; therefore, concurrent use of specific nutrients in combination with traditional cancer treatments may synergistically improve the overall care of GI cancer patients. This work aims to review and discuss the role of immunonutrition and microbiota modulation in improving nutritional status, postoperative recovery, and response to therapies in patients with GI cancer.

## 1. Introduction

Gastrointestinal (GI) cancers, a collective term encompassing a group of malignancies originating in various parts of the gastrointestinal tract, account for around 20% of newly diagnosed cancers and are responsible for over 25% of cancer-related deaths worldwide [1]. GI cancers differ in their potential to disrupt normal digestive functions and can have significant implications for a patient’s overall health and well-being, making early detection and appropriate treatment essential for optimal outcomes. The substantial global impact of GI cancers underscores the critical necessity to address the multifaceted challenges faced by these patients. Nutritional intervention and monitoring play pivotal roles in the comprehensive care of GI cancer patients. Managing the nutritional needs of these patients is essential not only to address the metabolic alterations associated with tumors but also to mitigate the adverse effects of cancer treatments. Adequate nutritional support can help optimize treatment outcomes, enhance the patient’s quality of life (QoL), and improve their overall prognosis. This review attempts to investigate the promising avenues of immunonutrition and microbiota modulation as innovative approaches to improve the nutritional status and overall well-being of GI cancer patients. By addressing these aspects, the goal is to offer insights into strategies that can complement traditional cancer treatments and enhance the holistic care of this cancer population.

## 2. Nutritional Status of GI Cancer Patients

In the realm of GI cancer care, understanding the nutritional status of patients is of paramount importance. Patients diagnosed with cancer frequently experience malnutrition as their tumors progress. The prevalence of malnutrition among cancer patients varies, ranging from 20% to 70%; in patients diagnosed with upper GI cancer, its prevalence can escalate to as high as 60.2% [2,3]. This malnutrition has been linked to increased postoperative complications and a decline in overall QoL. Malnutrition in cancer patients is a complex issue and can be classified into various categories, including anorexia, sarcopenia, and cachexia. These definitions are meant to help clinicians identify and treat metabolic and nutritional problems associated with cancer. Anorexia (loss of appetite) is quite common in cancer and, according to Abraham et al., 69% of patients newly diagnosed with gastric or gastroesophageal junction (GOJ) experience anorexia [4]. The primary cause of anorexia is often an increase in pro-inflammatory cytokines or an increase in lactate which can, in turn, modulate central nervous system neurotransmitter cascades [5,6]. Sarcopenia is characterized by diminished muscle strength, reduced muscle mass or quality, and decreased physical performance [7]. In GI cancer patients, among whom the prevalence of sarcopenia exceeds 40% [8], monitoring this condition is crucial as it significantly influences postoperative outcomes and is associated with a higher risk of complications [9,10]. Cancer cachexia, a syndrome characterized by loss of weight, muscle, and fat mass that cannot be reversed through conventional nutritional interventions, creates a proinflammatory environment leading to heightened energy expenditure [11]. The incidence of cancer cachexia in GI cancer patients varies widely, ranging from 15% in prostate cancer to as high as 90% in pancreatic cancer, with an overall occurrence between 40% and 80% [12,13]. GI cancer patients, in general, are at an elevated risk of experiencing cancer cachexia, often diagnosed too late for effective prevention or treatment of muscle and weight loss, resulting in heightened morbidity and mortality, reduced QoL, and suboptimal therapeutic outcomes [14].

Malnutrition in GI cancer patients has multiple underlying causes. The metabolic alterations induced by tumors and the adverse effects of cancer treatments can lead to a spectrum of nutritional challenges, including decreased appetite, difficulty swallowing, taste and smell changes, weight loss, fatigue, and a decline of QoL [2,15,16]. GI mucosa is sensitive to chemotherapy cytotoxicity: DNA damage, apoptosis, and inflammation of the healthy mucosa disrupt the GI architecture [17] by decreasing the mucosal area available for nutrient absorption and impairing the immune response [18]. This condition can be defined as gastrointestinal mucositis (GI-M) [19,20]. Consequently, patients who develop GI-M are at a significantly higher risk of secondary complications, especially nutritional deficiencies [21]. In addition, psychological distress and anxiety significantly influence dietary intake [22]. The connection between malnutrition and psychological distress has been evidenced in cancer patients. Those experiencing weight loss and other malnutrition-related symptoms often report heightened levels of psychological distress, manifested as increased fatigue, insomnia, heightened anxiety, and depression, ultimately exacerbating the progression of their illness [23]. When considered collectively, these factors can significantly impact the nutritional status of GI cancer patients and individuals who have undergone surgery as well as those currently undergoing radiotherapy and chemotherapy [24] (Figure 1). 

## 3. Nutritional Strategies for GI Cancers Patients Care

Clinical management guidelines emphasize the importance of screening and assessing malnutrition in cancer patients. They support the provision of adequate and effective nutritional therapy, particularly for people who face problems with nutrition [2,25]. While the precise timing for initiating nutrition support is not yet fully defined, it is recommended to commence intervention before malnutrition becomes well established [2]. In cases where patients are severely malnourished and actively undergoing treatment, nutritional support should be implemented immediately to address their nutritional needs. GI cancer patients undergoing radiotherapy can benefit from early and intensive nutritional intervention, which will improve their nutritional status and QoL [26,27]. Nutritional therapy in GI patients should, at first, comprise an adequate calorie and protein intake, essential to preserve lean body mass, promote wound healing, and support the body’s immune function. High-calorie, high-protein foods should be included in the diet, along with small, frequent meals [28,29]. Secondly, it is important to manage digestive symptoms, since GI cancers and their post-surgical treatments can lead to nausea, vomiting, diarrhea, constipation, and loss of appetite [30].

The dose of nutritional therapy is based on the patient’s energy and nutrient requirements. The recommended energy intake for cancer patients is 25–30 kcal/kg/day. Protein intake should be above 1 g/kg/day and above 1.5 g/kg/day if possible [15,31].

Effectively managing symptoms is a crucial aspect of cancer care from the time of diagnosis to treatment. These factors not only impact the well-being of GI cancer patients but also have substantial implications for treatment outcomes. 

Avoiding spicy or greasy foods, eating smaller portions, and consuming more frequent meals of easily digestible foods are dietary modifications that can help relieve these symptoms and improve nutrient absorption. Vomiting or diarrhea may also result in fluid loss; therefore, adequate hydration to support overall bodily functions should be implemented. Drinking water, clear broths, herbal teas, and consuming hydrating foods like fruits and vegetables can help meet hydration needs [32]. Certain micronutrients may be compromised in GI cancer patients due to reduced intake or absorption. Then, the healthcare team may recommend specific supplements to address deficiencies, such as vitamins (e.g., vitamin D, B vitamins) and minerals (e.g., iron, zinc), based on individual needs [33,34] (Figure 2). At last, in cases where oral food intake is insufficient or not possible, artificial nutritional may be required [35]. This includes enteral nutrition (EN) (delivery of nutrients through the GI tract, either orally or via a feeding tube), parenteral nutrition (PN) (delivery of nutrients directly into the bloodstream), nutrition counseling, and oral nutritional supplements (ONS). If the patient can eat but is malnourished or at risk of malnutrition, interventions to increase food intake or ONS are recommended. If the patient is undernourished, medical nutrition (EN or PN) is indicated [15]. After hospitalization, or when palliation is the main purpose of nutritional intervention, EN should be preferred, except when intestinal obstruction, ileus, severe shock, intestinal ischemia, high-flow fistula, or severe intestinal bleeding occurs [10]. Optimal preoperative nutritional support for at least 10 days has been reported to reduce the risk of postoperative surgical site infection (SSI) in patients with gastric cancer [36]. Clinicians should plan individualized nutritional interventions during screening, evaluation, and treatment processes and should not hesitate to prefer the EN and/or PN route when nutritional goals cannot be achieved with an oral diet alone [37] (Figure 2).

Preoperative nutrition and exercise intervention have been reported to provide perioperative functional improvement in esophagogastric cancer surgery patients [38]. According to the ESPEN guideline, routine postoperative nutritional support should be considered for surgical cancer patients (especially those undergoing upper GI cancer surgery) at moderate or severe nutritional risk [15]. Postponing a regular oral diet after major surgery is associated with an increased rate of infectious complications and a longer recovery [39].

Several studies have highlighted the critical role of adequate nutrition in mitigating postoperative complications, maintaining immune function, reducing treatment toxicity, enhancing overall survival rates, shortening length of stay (LOS) in hospital, and facilitating the timely administration of adjuvant oncologic therapy [40,41,42,43,44].

Following surgery, colorectal cancer (CRC) patients receiving oral nutritional supplements (ONS) demonstrated notable enhancements; however, no significant alterations were observed in metrics like body weight and body mass index (BMI) [45]. Nevertheless, a randomized trial showed that the use of ONS administration after GI cancer surgery may have positive outcomes on patients’ body weight and BMI [46]. Likewise, patients administered with ONS three months following GC surgery experienced significantly less reduction in body weight and exhibited a notably higher BMI compared with patients who received nutrition counseling alone [45]. A prospective randomized controlled study showed that patients experienced a decline in nutritional status after discharge, highlighting the critical role of postoperative nutritional supplementation in enhancing nutritional status, QoL, and reducing morbidity among surgical patients [47]. The European Society for Clinical Nutrition and Metabolism (ESPEN) practice guideline recommends an enhanced recovery after surgery (ERAS) program for all cancer patients undergoing surgery [15]. ERAS is an evidence-based, multicomponent perioperative protocol that aims at reducing stress and promoting a return to function [48]. Within the scope of this program, each patient should be assessed for malnutrition, and additional nutritional support should be applied if necessary. Nutritional components of ERAS are avoiding fasting, preoperative fluid, and carbohydrate overload, and recommencement of oral diet on the first postoperative day. The aim is to minimize the metabolic response to surgery [15].

Considering all these data, it becomes crucial that each step of the process for GI cancer surgery patients, starting from preoperative nutritional screening and extending to post-discharge nutritional support, is overseen by a healthcare professional. Adequate nutritional care should be meticulously planned for each patient (Figure 2).

However, long-term prospective studies of GI cancer patients’ preoperative and postoperative nutritional care in larger populations are necessary.

## 4. Immunonutrition and Gut Microbiota Modulation

Cancer and its treatment can weaken the immune system, making patients more susceptible to infections and impairing their ability to combat the disease. Moreover, GI surgical procedures trigger a natural inflammatory response in the body, which is intended to be protective and create an environment conducive to recovery. This response primarily aims to facilitate energy production, restore cardiovascular balance, promote tissue repair and wound healing, and ultimately ensure the state of well-being of the patient. However, there are instances when this inflammatory response becomes dysregulated, leading to the release of proinflammatory cytokines, endothelial dysfunction, glycocalyx damage, activation of neutrophils, and subsequent damage to tissues and multiple organ systems [49]. In such cases, patients undergoing GI surgery may experience postoperative complications such as anastomotic dehiscence and surgical site infections, or even face severe consequences due to an amplified and uncontrolled inflammatory reaction [50].

Immunonutrition in GI cancer care is designed for complementing traditional cancer treatments by supporting the immune system, boosting the host’s cancer-related immune response, reducing inflammation, promoting tissue repair, and improving overall nutritional status [51].

Interestingly, inflammation is closely linked to alterations in the gut microbiota (GM) and their metabolites, particularly short-chain fatty acids (SCFAs). As a counterbalance to the immune response, the human GM appears to play a significant role in the development of post-surgery complications [52]. Furthermore, the GM is integral in the absorption, storage, and utilization of energy derived from dietary intake [53]. It contributes to food regulation intake by influencing hormones related to metabolic function and brain regions associated with eating behavior [54]. In addition, it has beeen proven that the GM plays critical roles in protecting the integrity of the GI mucosa and the maintenance of its homeostasis [55]. GI cancers and also their treatments, such as chemotherapy, have been consistently demonstrated to induce changes in the GM composition and functions [56], contributing to GI-M severity and to malnutrition onset [57,58]. In this context, dietary interventions aimed at fostering a healthy GM before therapy and enhancing its resilience during and after therapy show potential as therapeutic strategies for GI-M and its related symptoms [59,60].

The maintenance of a functional gut barrier and of host homeostasis is granted by the action of beneficial bacteria like *Bifidobacterium* spp., *Faecalibacterium prausnitzii*, *Lactobacillus* spp., and the production of SCFAs, particularly butyrate. SCFAs stimulate the regeneration of epithelial cells, the production of mucus and antimicrobial peptides, and modulate T regulatory (Treg) cells [61,62]. Dendritic cells located in the lamina propria release transforming growth factor-β (TGF-β) in response to commensal antigens, activating Treg cells to secrete interleukin-10 (IL-10) and TGF-β, promoting a more immune-tolerant phenotype [63]. In addition, dietary and microbiota-derived ligands of the aryl hydrocarbon receptor (AhR) stimulate innate lymphoid cell 3 (ILC3) to produce IL-22, which plays a vital role in preserving intestinal barrier function [64]. Another hallmark of intestinal balance is a thicker mucus layer, acting as a barrier between luminal bacteria and epithelial cells. 

When dysbiosis occurs, the lower abundance of beneficial bacteria and higher abundance of pathobionts (i.e., *Clostridium difficile* and *Escherichia coli*) promote production of inflammatory factors such as radical oxygen species (ROS), nitric oxide (NO), proinflammatory cytokines, and cyclo-oxygenase 2 (COX-2) [65,66]. The decrease in thickness of the mucus layer and cellular tight junction expression results in a compromised intestinal barrier function, allowing bacterial products, like lipopolysaccharides (LPSs), to leak from the intestinal lumen into the lamina propria. LPSs bind to toll-like receptors (TLRs), triggering macrophages to generate tumor necrosis factor-α (TNF-α). TNF-α promotes the proliferation of T helper cell type 1 (Th1) and the release of pro-inflammatory cytokines, including TNF-α and interferon-γ (IFN-γ), ultimately causing inflammation. This inflammatory process further undermines the integrity of the intestinal barrier. Additionally, the reduction in IL-10-producing Treg cells contributes to the inflammation within the intestine [67]. Finally, dysbiosis may increase colonic epithelial cells’ exposure to carcinogens [68]. Notably, the GM influences the host’s response to cancer therapy. Germ-free and antibiotic-treated mice showed reduced responses to immunotherapy and chemotherapy by CpG oligonucleotides, due to impaired function of myeloid-derived cells in the tumor microenvironment [69]. On the other hand, *Barnesiella intestinihominis* has been reported to have an adjuvant effect on cyclophosphamide (CTX)-induced tumor immunity by promoting infiltration of IFN-γ-producing γδT cells in cancer lesions [70]. 

Finally, studies report that the GM is involved in the management of cancer, as the composition of the GM can modulate the effect of anticancer drugs. Neoadjuvant chemoradiotherapy (nCRT) has become a standard treatment for locally advanced rectal cancer (LARC), with only 15–27% of patients achieving a pathological complete response and 20–40% achieving little to no response. Aiming to elucidate the mechanism underlying the response of LARC to nCRT, Teng et al. showed that GM-mediated nucleotide synthesis can modulate the response of LARC patients to nCRT. Multi-omics data integration showed that *Bacteroides vulgatus*-mediated nucleotide biosynthesis was associated with nCRT resistance in LARC patients, and non-responsive tumors were characterized by up-regulation of genes related to DNA repair and nucleoside transport [71].

Given these results, a more balanced microbiota obtained through microbiota manipulation may have positive contributions in preventing cancer formation and increasing response to medical treatment [72,73]. Therefore, managing the immune response during the post-surgical period, both in the short and long term, with a focus on the interaction between GM and inflammation (a bidirectional signaling axis regulating immune response, GI balance, and body weight through appetite control, energy storage, and expenditure), presents an advantageous strategy. 

### 4.1. Immunonutrients Supplementation

Among nutritional interventions, immune-enhancing nutrient formulas, also called immunonutrients, can be supplemented via EN, PN, and ONS. The most common immunonutrients are arginine, glutamine (Gln), omega-3 (ω-3) fatty acids, nucleotides, or RNA, which have been seen to modulate inflammatory responses and increase protein synthesis following surgical procedures [74,75]. The advantages obtained from immunonutrient consumption encompass various mechanisms, among them GM modulation. Studies indicate that arginine treatment in mice results in a beneficial alteration of the Firmicutes-to-Bacteroidetes ratio to favor Bacteroidetes, along with decreased expression of nuclear factor-κB (NF-κB), mitogen-activated protein kinase (MAPK), and phosphatidyl inositol 3-kinase/protein kinase B (PI3K/Akt) signaling pathways [76]. Notably, Bacteroidetes promote intestinal innate and mediated immunity, including the secretion of immunoglobulin A (IgA) and various cytokines [76,77]. These findings are particularly significant in the context of diseases like Crohn’s disease (CD) and ulcerative colitis (UC), where Bacteroidetes’ presence is diminished, potentially contributing to the reduced anti-inflammatory effect observed during colitis [78,79]. Similarly, ω-3 fatty acids can modulate the abundance of gut microorganisms. Recent studies suggest that dietary supplementation with ω-3 polyunsaturated fatty acids increases the abundance of various health-promoting bacteria, including butyrate producers from genera such as *Bifidobacterium*, *Roseburia*, *Lactobacillus*, and the mucin specialist *Akkermansia muciniphila* [80,81,82]. Interestingly, in a cross-sectional study of breast cancer survivors, higher blood levels of docosahexaenoic acid (DHA) were positively associated with an increased abundance of *Bifidobacterium* in the GM, particularly in participants without a history of chemotherapy [83]. Nevertheless, the influence of immunonutrient use on microbiota is still inadequately investigated, highlighting the need for further research to comprehensively understand the impact of immunonutrition in prevent dysbiosis, especially in the context of GI cancer care. According to the ESPEN guidelines, oral or enteral immunonutrient administration is recommended for upper GI cancer surgery patients in preoperative and postoperative nutritional care [15]. However, more research is needed to assess the efficacy of immunonutrient-enriched formula supplementation compared with standard oral and enteral nutrition in the perioperative period [10]. In a recent meta-analysis, enteral immunonutrition was found to be both safe and effective in reducing overall complications, particularly infectious complications, and it also led to a shortened hospital stay. This positive outcome was observed in patients undergoing surgery for GI cancers, including GC, CRC, esophageal cancer, periampullary cancer, and pancreatic cancer [84].

In studies involving patients undergoing laparoscopic colorectal resection, preoperative and postoperative immunonutrient supplementation was associated with a lower incidence of SSI compared with those who received nutrition counseling alone [85]. Similarly, preoperative EN immunonutrition has been proven effective in preventing SSI in CRC patients without malnutrition [86].

However, when comparing standard ONS with ω-3-enriched ONS in CRC surgery patients, ω-3-enriched ONS did not significantly affect postoperative complications, LOS, postoperative blood loss, the need for intensive care, or hospital readmission [87].

On the other hand, Adiamah et al. reported a 48% risk reduction of postoperative infectious complications in patients receiving preoperative immunonutrition. This intervention also led to shortened LOS of 1.5 days, although it did not impact other complications or mortality [40].

Similarly, Probst et al. found that perioperative immunonutrition reduced infectious complications, general complications, and LOS, with no effect on mortality [88]. Furthermore, a pilot trial assessing perioperative nutritional supplementation in GI cancer patients demonstrated a feasible enrollment fraction of 49% and revealed a higher proportion of infectious complications in the control group, emphasizing the importance of infectious complications as a relevant outcome of interest in such studies [89]. 

Furthermore, a retrospective study by Franceschilli et al. suggested that the combination of preoperative immunonutrition within the context of the ERAS protocol for patients with normal nutritional status undergoing laparoscopic total gastrectomy (LTG) reduced postoperative complications [90].

In the case of malnourished patients, enteral immunonutrition affected postoperative complications and LOS. However, for patients without malnutrition, the content of nutritional support did not significantly impact complications or LOS [91].

Studies have shown that early postoperative enteral immunonutrition enriched with nutrients like arginine, ω-3 fatty acids, and RNA positively influenced surgical wound healing and immune function in patients undergoing gastrectomy for GC [92,93].

Additionally, immunonutrition has been reported as a safe and feasible nutritional therapy that positively modulates immune responses after esophagectomy [93]. However, in another randomized controlled trial, no significant immunomodulatory effect was observed when comparing immunonutrient-rich EN with standard EN [94].

Lastly, in GI cancer surgery patients, the switch from standard intravenous fluid to immune-enhancing EN reduced infectious complications by two-thirds, while non-infectious complications saw a 13.5% reduction. This result suggests that a nutritional intervention modulating the host immune response may positively influence the relationship between immune support and postoperative infections [95]. PN administration of immunonutrients gave similar results. In an interventional clinical study, GI cancer surgery patients were assigned to two different groups: one group received postoperative total parenteral nutrition (TPN), and the other group received TPN along with a daily supplementation of 0.4 g/kg of Gln. Following these interventions, the nutritional status improved in both groups; however, the group receiving supplementation exhibited significantly greater improvement and demonstrated better results in GI function assessment [96]. Lu et al. previously demonstrated that Gln-enriched TPN led to higher serum prealbumin levels, improved nitrogen balance, and lower levels of inflammatory markers such as IL-6 and C-reactive protein (CRP) compared with standard TPN in postoperative GI cancer patients [97]. These results suggest that Gln-enriched TPN may enhance both nutritional and inflammatory status and potentially reduce the risk of infectious complications in these patients. Regarding the roles of immunonutrition in modulating radio- and chemotherapy side effects, Gln may shorten the duration of chemotherapy-induced diarrhea but does not affect its severity [98]. A systematic review evaluating Gln intake among colon and CRC patients found that Gln may reduce some chemotherapy-induced complications, such as GI-M and diarrhea, and improve postoperative nitrogen balance, immunity, and wound healing, whereas Gln had no beneficial effects on the side effects of radio-chemotherapy [99]. On the other hand, long-chain ω-3 fatty acids and fish oil are recommended to improve body weight, food intake, and other components in patients undergoing chemotherapy and at risk of weight loss or malnutrition [15]. 

Anyway, the current evidence does not provide a clear role for immunonutrition in managing infectious episodes during chemotherapy in cancer patients [100]. Interestingly, ω-3 has shown potential to enhance the effectiveness of chemotherapy through its synergistic inhibition of cell growth [101]. Mechanistic insights into ω-3’s action were gained through in vitro studies conducted on CRC cell lines, revealing its antiproliferative effects [102,103], promotion of apoptosis [101,104], and improved chemotherapy efficacy [101,102].

The effect of eicosapentaenoic acid (EPA) supplementation in GI patients has been studied with varying outcomes. In a double-blind, placebo-controlled study involving advanced cancer patients with weight and appetite loss, daily administration of 1.8 g of EPA C20:5 ω-3 for two weeks did not result in significant improvements in appetite, fatigue, nausea, overall well-being, caloric intake, nutritional status, or functional abilities compared with the placebo group [105]. A clinical trial reported that dietary counseling by qualified dietitians and the use of EPA-ONS in advanced CRC patients receiving chemotherapy could help maintain weight and potentially enhance symptom control, nutritional status, and QoL [106]. A recent study assessed the impact of perioperative EPA supplementation in patients with localized gastric cancer, as part of a randomized clinical trial. The study found that, overall, there was no significant survival benefit associated with perioperative EPA. However, subgroup analyses indicated potential benefits in patients who received neoadjuvant chemotherapy (NAC) and those with nodal metastasis [107]. Further research may be needed to clarify the specific patient populations that could benefit from EPA supplementation. All these results suggest that immunonutrition can help GI cancer patients in many ways, from perioperative care to symptom reduction and immune system support (Figure 3).

### 4.2. Calorie Restriction and Fasting

Calorie restriction (CR) is recognized for its anti-inflammatory effects mediated by various mechanisms, demonstrating a beneficial influence on the prevention and treatment of conditions characterized by hyper-inflammatory responses. CR is a nutritional intervention that restricts energy intake by 25–30% without causing malnutrition or deprivation of essential nutrients. Since most dietary energy comes from carbohydrates, energy restriction indirectly leads to carbohydrate restriction. Accordingly, calorie restriction is assumed to regulate effector immune activities that use glucose as the primary substrate [108]. Given that glucose is the primary fuel for cancer cells, calorie restriction emerges as a promising nutritional therapy for individuals with cancer. CR holds potential in exerting anticancer effects by triggering molecular pathways that enhance cellular defenses, support DNA repair, and mitigate oxidative damage [109]. An important aspect of calorie restriction during cancer treatments is fasting, which prompts distinct responses in cancer cells compared with normal cells [110]. In normal cells, fasting leads to a reduction of proteins and enzymes related to cell growth, such as insulin-like growth factor 1 (IGF-1), mammalian target of rapamycin (mTOR), protein kinase A (PKA), and protein kinase B (PKB/AKT). This outcome induces growth arrest or reduction in healthy cells, promoting cell survival and enhancing cellular protection against chemotherapeutic agents—a phenomenon also referred to as differential stress resistance (DSR). In cancer cells, fasting induces differential stress sensitization (DSS), rendering them more susceptible to chemotherapeutic agents and promoting increased cell death [111]. The dual impact of fasting-induced autophagy in cancer underscores its potential diverse applications in cancer treatment. CR holds promise in enhancing treatment effectiveness by modulating autophagy and preserving normal cells. Studies have demonstrated that combining autophagy inhibition with calorie restriction reduces tumor growth more effectively than individual treatments [111,112]. Fasting is suggested as a procedure to halt cancer development and tumor growth by suppressing pathways that activate tumor growth and activating pathways that inhibit tumor growth in tumor cells, thus preventing disease progression [113,114]. However, it is crucial to emphasize that these outcomes were observed with prolonged fasting (>48 h). Additionally, research indicates that short-term fasting can sensitize cancer cells to chemotherapeutic agents, enhancing the efficacy of radiation and chemotherapy [115,116]. Clinical studies involving cancer patients undergoing chemotherapy have demonstrated that fasting, even in the short term, is safe and well tolerated, potentially improving treatment outcomes and enhancing QoL, although in some cases, fasting may have no notable effect [117,118,119].

Some of the positive effects of CR and fasting may be related to the impact of diet on adult stem-cell function [120]. CR enhances intestinal stem cells (ISCs) and neighboring niche cell numbers and increases stem cells’ self-renewal capacity in response to reduced mTOR signaling from Paneth cells [121]. CR decreases PI3K/AKT signaling pathways by reducing circulating insulin/IGF-1 levels and suppresses cell survival in a colon-derived human cancer cell line (SW620), accompanied by increased expression levels of forkhead box O (FOXO) target genes [121]. In addition, CR inhibits colon tumor cell (MC38) growth by regulating NF-κB activation and inflammation-related gene expression [122]. Similarly, fasting induces ISC self-renewal, mediated by peroxisome proliferator-activated receptor δ (PPARδ) triggered by the oxidation of free fatty acids released from adipose tissue. This depends on the nutrient-sensing capacity of the ISCs [120]. Further, Deng et al. showed that fasting reduced leptin-receptor-positive (Lepr+) cell numbers and, thus, serum leptin levels. This leads to a decrease in insulin-like growth factor 1 (Igf1) secreted by Lepr+ cells. As a result, the proliferation of ISCs and progenitor cells is reduced during fasting. It is noteworthy that Lepr+ mesenchymal cells (MCs) perceive dietary changes. However, no apoptotic cells were detected in MCs of intestinal crypts after fasting, indicating that Lepr+ cells decrease independently of apoptosis [123]. Unlike other stem cells, intestinal stem cells coexist with the intestinal microbiota population but live separately in their own integrity. Therefore, the relationship between the microbiota and ISCs needs to be considered. Gut microbes can be devastating, given the vital role of the long-term integrity and functionality of ISCs and progenitor cells. By causing biological damage, the GM promotes the regeneration of the epithelial layer. This leads to defense against pathogens and immunomodulatory effects [124]. Interestingly, fasting has the potential to exert immunomodulatory effects by modulating the microbiome. In mice, it was demonstrated that restricting caloric intake led to GM alterations, specifically an increase in *Lactobacillus* spp., believed to offer protection against invading pathogens and to lower inflammatory cytokine levels, and a decrease in Streptococcacae, known inducers of mild inflammation [125]. Additional probiotic treatment could further amplify the beneficial effects of fasting. In a pilot study involving overweight individuals, a 1-week fasting diet followed by a 6-week probiotic intervention resulted in increased GM diversity and abundance of mucin-degrading bacteria, notably *Akkermansia muciniphila*, with the probiotic formula bolstering specific administered gut microbial populations [126]. 

Although the effects of calorie restriction and fasting on cancer are promising, cancer-related clinical conditions such as malnutrition, cachexia, a possibly weakened immune system, and susceptibility to infection should be taken into account when evaluating the effectiveness of long-term fasting interventions alone in cancer treatment [110]. Therefore, to avoid adverse effects on immune function, it is crucial to implement caloric restriction and fasting interventions in cancer patients in a controlled manner and to maintain balance by maintaining adequate caloric intake.

### 4.3. The Role of Biotics

One promising avenue for GI cancer patients care is the modulation of the GM using biotics (probiotics, prebiotics, and synbiotics) (Figure 3). Specific probiotic strains of *Bifidobacteria*, *Lactobacilli*, *E. coli*, *Propionibacterium*, *Bacillus*, and *Saccharomyces* can beneficially modulate TLR activation by reducing the activation of MAPK and NF-κB pathways and the production of pro-inflammatory cytokines [127]. It is assumed that the *Bifidobacterium longum* subsp. *longum GT15* strain aims to maintain normal healthy functions by responding to pro-inflammatory cytokines. In addition, overexpression of heat shock protein 20 (Hsp20), which is known to play a role in reducing inflammation, reduces TNF-α expression. In the *B. longum* subsp. *longum GT15* strain exposed to TNF-α, the transcription of the BLGT_*RS00625* gene encoding for Hsp20 increased five-fold, suggesting that this mechanism may be one of the pathways used by *Bifidobacteria* to reduce inflammation [128]. 

Moreover, a distinct group of Gram-positive bacteria can predominantly produce bacteriocins, a group of bacterial peptides, which display antimicrobial activity against other bacteria [129]. Bacteriocin producers include various genera, such as *Pediococcus*, *Leuconostoc*, *Lactococcus*, *Enterococcus*, *Streptococcus*, *Lactobacillus*, and *Bifidobacterium* [130,131]. Bacteriocins selectively target pathogens, but not commensal GM [132], and exert cytotoxic activity against cancer cells [133,134]. Intriguingly, some bacteriocins exhibit immunomodulatory properties, thus participating in the maintenance of a balanced crosstalk between GM and immunity [135]. Indeed, bacteriocins secreted by *Bacillus subtilis* were described as stimulators of innate immune response via IL-1β, IL-6, TNF-α, and NO production in both in vitro cells and mouse peritoneal macrophages [136]. The enhanced phagocytosis of macrophages correlated with the TLR4 and the NF-κB and MAPK signaling pathways [137]. Treatment of human peripheral blood mononuclear cells (PBMCs) with acidocin A, a bacteriocin, resulted in increased production of multiple cytokines and chemokines, including macrophage inflammatory protein (MIP)-1α, MIP-1β, IL-6, and TNF-α [138]. Moreover, *Lactobacillus plantarum* genes encoding production or secretion of bacteriocins were reported to enhance production of IL-10 over IL-12 and TNF-α induction in dendritic cells (DCs) and in PBMCs [139,140].

Other immune-modulatory functions exerted by the commensal *Bacteroides fragilis* are the secretion of polysaccharide A that is recognized by the heterodimer TLR2/TLR1 in cooperation with Dectin-1 and induces the cAMP response element-binding protein (CREB)-dependent expression of anti-inflammatory genes [141]. *Bacteroides fragilis* can also suppress the Th17 responses by promoting Tregs through TLR2 signaling [142]. Also, *Lactobacillus reuteri*, *Lactobacillus murinus*, and *Helicobacter hepaticus* can increase the proportion of IL-10 producing Tregs in mice [143,144,145,146,147]. 

A randomized, double-blind, placebo-controlled trial in CRC patients undergoing colon–rectal resection demonstrated the benefits of probiotics. Patients received a combination of *Lactobacillus* and *Bifidobacteria* strains, which included *Lactobacillus acidophilus*, *Lactobacillus lactis*, *Lactobacillus casei* spp., *Bifidobacterium longum*, *Bifidobacterium bifidum*, and *Bifidobacterium infantis* twice a day for six months. This intervention led to a decrease in pro-inflammatory cytokines and a significant reduction in post-surgical complications [148]. Another randomized controlled prospective study involving CRC patients administered a compound of eight bacterial cultures, including various *Lactobacillus* and *Bifidobacterium* strains, showed promising results. Treated patients exhibited a lower frequency of post-surgical complications, reduced operative and postoperative LOS, and a lower mortality rate in a six-month postoperative follow-up compared with untreated patients [149]. In a recent study, 100 CRC patients receiving supplementation with a probiotic containing *Bifidobacterium infants*, *Lactobacillus acidophilus*, *Enterococcus faecalis*, and *Bacillus cereus* showed dysbiosis alleviation and increased production of SCFA compared with controls [150]. A quantitative meta-analysis involving 14 studies and 1566 patients demonstrated a significant benefit of probiotics and synbiotics administration in both pre-and post-surgical care, especially reducing postoperative infections [151]. Moreover, no significant results were reported in randomized, double-blind controlled studies on patients undergoing ileostomy, performed to prevent further damage associated with anastomotic leak in CRC patients, with perioperative administration of *Lactobacillus plantarum CJLP243* [152], or of a probiotic mix (*Lactobacillus acidophilus DSM 24735*, *Lactobacillus acidophilus DSM 24735*, *Lactobacillus plantarum DSM 24730*, *Lactobacillus plantarum DSM 24730*, *Lactobacillus paracasei DSM 24733*, *Lactobacillus paracasei DSM 24733*, *Lactobacillus delbrueckii* subsp. *bulgaricus DSM 2*, *Bifidobacterium breve DSM 24732*, *Bifidobacterium longum DSM 24736 113*, *Bifidobacterium longum DSM 24736*, *Bifidobacterium infantis DSM 24737*, *Streptococcus thermophilus DSM 24731*) [153]. More studies are needed to assess the efficacy of the treatment and subsequent inclusion of probiotics administration in a protocol before ileostomy [152]. Similar findings were observed in gastric adenocarcinoma patients undergoing radical gastrectomy, where probiotics administration reduced levels of inflammatory markers [154]. 

GI surgery includes several other interventions, such as hepatectomy with extrahepatic bile duct resection, esophagectomy, and pancreatoduodenectomy, which may result in bacterial translocation to mesenteric lymph nodes (MLNs) and from there to the bloodstream. Since the presence of bacteria in MLNs is directly associated with postoperative infections, it has been observed how a pre-operative administration of synbiotics can improve the intestinal microenvironment and prevent postoperative infections in esophagectomy [155]. Hepatectomy that implies extrahepatic bile duct resection and pancreatoduodenectomy is discussed below [156].

In esophageal cancer, where NAC is recommended as a standard treatment before surgery [157], it has been observed in a randomized control trial that the co-administration of synbiotics reduces the toxicity provoked by the chemotherapy treatment and prepares the intestinal environment for highly invasive surgery, with lower bacterial translocation to the MLN and to the bloodstream [158]. 

In more severe conditions, patients with advanced stages of esophageal cancer are also treated with prophylactic antibiotics during NAC. In a multicenter randomized study, patients undergoing a pre-operative cycle of NAC were enrolled and randomly assigned either antibiotic administration or a symbiotic administration combined with EN. In this trial, synbiotics administration with EN diminished the side effects of the chemotherapy on the intestinal tract, such as diarrhea, and resulted in an alternative treatment to antibiotics [159]. 

Unfortunately, synbiotics administration is not beneficial for all patients, and detecting microbiota species in patients’ guts prior to chemotherapy may also be predictive of the efficacy of a co-adjuvant synbiotics treatment [160]. As tested by Sugimoto et al. in a retrospective exploratory study, *Anaerostipes hadrus* and *B. pseudocatenulatum* may mitigate chemotherapy side effects and allow the protective role of synbiotics + EN administration during NAC [160]. Considering malignant hepatic neoplasms, hepatic resection is a standard treatment procedure in many cases, but the mortality rate of the surgery is still moderately high, about 3.5% [161]. A meta-analysis of a total of four studies involving 205 patients assessed that the pre-operative administration of prebiotics reduced postoperative infections and the need for antibiotics [162]. However, in a more recent randomized controlled trial, patients with resectable hepatocellular carcinoma administered with probiotics prior to the resection showed no beneficial results regarding bacteria translocation or post-surgical infections [163]. Finally, in pancreatic cancer patients, where the intestinal dysbiosis has been documented [164], the use of pre and probiotics could be a therapeutic approach to alleviate side effects of chemotherapy/radiotherapy, but no significant results have yet been reported. For instance, a trial involving the administration of M-20 (a biotherapeutic agent of soybean fermentation metabolites and microorganisms that reproduce the intestinal environment) to prevent cachexia in pancreatic cancer patients treated with chemotherapy is at its early stage of recruiting (NCT04600154).

While the use of probiotics is generally safe and well tolerated in the general population, their application in vulnerable subpopulations requires careful consideration of several factors, including a thoughtful probiotic selection. For this purpose, the safety and efficacy of diverse formulations used as adjunctive probiotics in oncological surgery have been assessed in various studies [165,166,167,168]. In a systematic review, Cogo et al. evaluated 21 different probiotics formulations in oncological surgery, within 36 randomized controlled trials involving 3305 participants and six nonrandomized/observational cohort studies [165]. Their findings support the belief that the effects of probiotics are specific to the product and formulation, with the most promising results obtained with the post-surgery oral supplementation of *Lactobacillus acidophilus* LA-5 + *Lactobacillus plantarum* + *Bifidobacterium lactis BB-12* + *Saccharomyces boulardii* in CRC patients. With regards to safety, among the randomized controlled trials, 47% of patients did not furnish specific data on side effects, 25% did not experience adverse events, while 28% reported common side effects being mild and encompassing nausea and flatulence, indicating a favorable safety profile. However, a small proportion of patients (6%) reported elevated rates of specific complications with the probiotics arms, including pancreatic fistula and 30-day readmission [165]. These findings underscore the necessity for a cautious and ongoing evaluation of the safety profile of probiotics, especially in the context of surgical interventions for individuals with cancer.

## 5. Conclusions

In light of the intricate and multifaceted factors contributing to nutritional imbalances in GI cancer patients, effective treatments necessitate a comprehensive and multi-disciplinary approach. Immunonutrition and microbiota modulation emerge as promising avenues to enhance nutritional status, regulate immune response, promote tissue repair, and modulate the side effects of anticancer drugs. Bridging the nutritional gap by employing immunonutrients and microbiota modulators early during disease onset can help stabilize weight loss, enhance treatment tolerability, reduce the decline in performance status, prevent infections, and improve survival rates. Unfortunately, clinical practice often involves late-stage assessments, where multiple nutritional deficiencies have already surfaced, and cancer cachexia has become resistant to conventional treatments, potentially yielding contradictory results. Therefore, it is imperative to acknowledge the need for more rigorous clinical trials to thoroughly assess the impact of these interventions.

## Figures and Tables

**Figure 1 nutrients-15-04408-f001:**
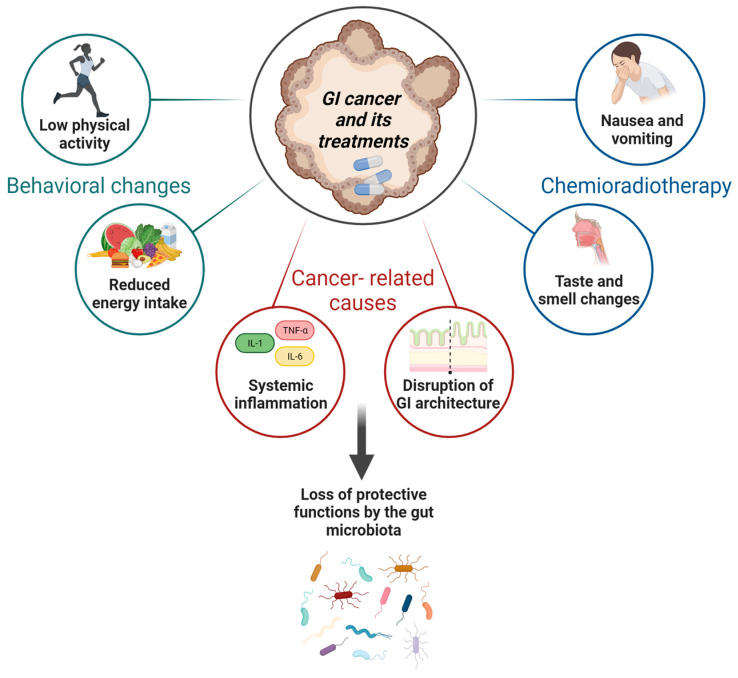
Underlying causes of malnutrition in GI cancers. Nutritional alterations are due to various factors: reduction in energy intake and poor physical activity; consequences of chemoradiotherapy such as nausea, vomiting, taste changes, and cancer-related causes such as systemic inflammation and difficulty absorbing nutrients (due to destruction of the GI architecture). Alterations in the GM composition may be attributed to all three factors described (image created with Biorender.com).

**Figure 2 nutrients-15-04408-f002:**
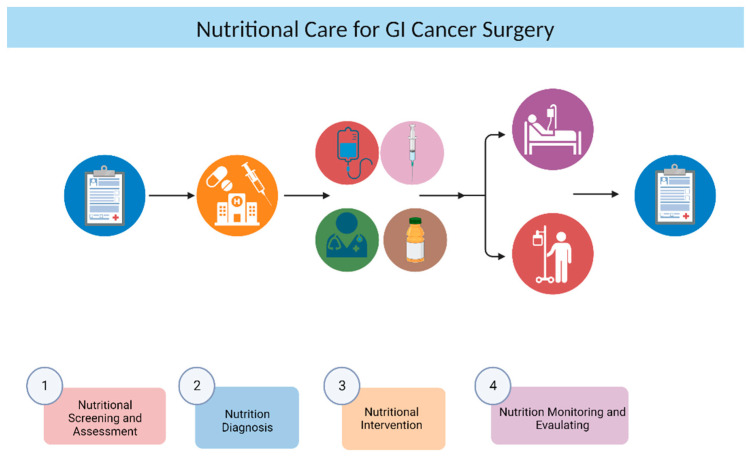
Nutritional Care Workflow for GI Cancer Surgery Patients. (1) The first step in planning nutritional care for surgical cancer patients involves screening and assessment of nutritional status. (2) This is followed by a nutrition diagnosis that commonly includes identifying conditions such as malnutrition, cachexia, or sarcopenia in cancer patients. (3) At least, the appropriate nutrition therapy (enteral nutrition, parental nutrition, oral nutritional supplement, or nutrition counseling) should be determined. (4) Postoperatively and during the discharge period, patients are closely monitored, and it is beneficial to reevaluate their nutritional status (image created with Biorender.com).

**Figure 3 nutrients-15-04408-f003:**
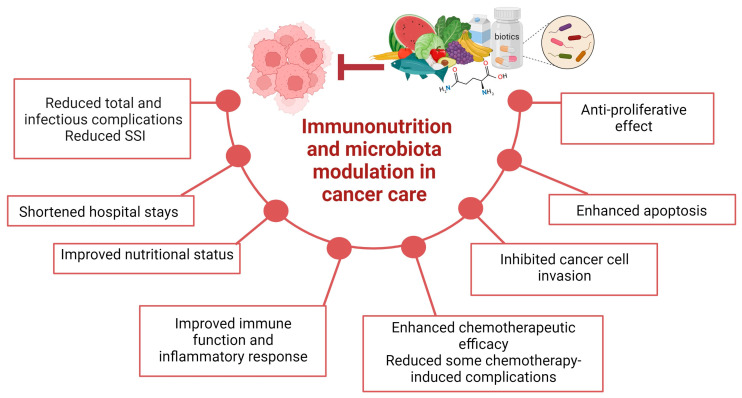
Overview of the potential benefits of immunonutrition and microbiota modulation in gastrointestinal cancer patient care. The combined effect of nutritional intervention and gut microbiota modulation can comprehensively improve patient care by reducing infection complications, shortening the hospital stay, improving the overall nutritional status, ameliorating the host immune response, and enhancing the effects of conventional anticancer treatments. At the intracellular level, an antiproliferative effect, an increase in apoptosis, and the inhibition of cancer invasion after nutritional and microbiota modulation interventions were described (image created with Biorender.com).

## Data Availability

No new data were created or analyzed in this study. Data sharing is not applicable to this article.

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
