# Peer review of "Synergistic Strategies for Gastrointestinal Cancer Care: Unveiling the Benefits of Immunonutrition and Microbiota Modulation"

_nutrients, 2023, doi:10.3390/nu15204408_

Round 1

Reviewer 1 Report

In this manuscript, Serena Martinelli and colleagues reviewed the benefits of the concurrent use of specific nutrients and microbiota modulation in combination with traditional cancer treatments may synergistically improve the overall GI cancer patient care, which provides a new therapeutic strategy for GI cancer. I found this study interesting and needed attention widely, yet for comprehensiveness, there are some critical issues that the authors must address. I detail my criticisms below:

1.      The author should include a discussion of the difference between malnutrition, fasting, and calorie restriction because the author claims that malnutrition can worsen the prognosis of the patients by affecting the proinflammatory environment, while according to previous studies, fasting also could affect the intestinal stem cell niche, reducing the abundance of mesenchymal, such as Lepr+ cells, which have a significant function to promote the proliferation of intestinal stem cells, and the relationship of intestinal stem cell proliferation and GI cancer is quite intimate.

2.      The author should give some specific examples of how gut microbiota and their metabolites regulate immune response. Such as how some specific strains result in pro-inflammatory cytokines and anti-inflammatory factors gene expression.

Author Response

In this manuscript, Serena Martinelli and colleagues reviewed the benefits of the concurrent use of specific nutrients and microbiota modulation in combination with traditional cancer treatments may synergistically improve the overall GI cancer patient care, which provides a new therapeutic strategy for GI cancer. I found this study interesting and needed attention widely, yet for comprehensiveness, there are some critical issues that the authors must address. I detail my criticisms below:

Point 1: The author should include a discussion of the difference between malnutrition, fasting, and calorie restriction because the author claims that malnutrition can worsen the prognosis of the patients by affecting the proinflammatory environment, while according to previous studies, fasting also could affect the intestinal stem cell niche, reducing the abundance of mesenchymal, such as Lepr+ cells, which have a significant function to promote the proliferation of intestinal stem cells, and the relationship of intestinal stem cell proliferation and GI cancer is quite intimate.

Reply 1: We thank the reviewer for raising this point and allowing us to implement our work. To address this point we added a new subsection “3.2 Calorie restriction and fasting”, including information regarding the mechanisms by which calorie restriction and fasting act on cancer metabolism.

Point 2: The author should give some specific examples of how gut microbiota and their metabolites regulate immune response. Such as how some specific strains result in pro-inflammatory cytokines and anti-inflammatory factors gene expression.

Reply 2: In agreement with the reviewer's suggestion, and for a better understanding of the importance of the microbiota functions, we added examples of how gut microbiota modulates the immune response in section 4 (Immunonutrition and Gut Microbiota Modulation) page 8, lines 254-282 and page 12, lines 497-532.

Author Response

Point 1: Assessment of the impact of probiotics and synbiotics: The review article mentions the administration of probiotics and synbiotics in gastrointestinal cancer patients and their potential to reduce toxicity, improve the intestinal microenvironment, and prevent postoperative infections. Including studies on the efficacy and safety of probiotics and synbiotics can provide valuable insights into their role in cancer treatment.

Reply 1: We thank the reviewer for the valuable comment. Accordingly, we implemented the subsection 4.2 “the role of biotics” with a paragraph concerning the safety and efficacy of probiotics and synbiotics in GI cancer care (lines 608-627).

Point 2: Recommendations for specific cancer types and surgical procedures: The authors briefly mention the impact of immunonutrition and microbiota modulation in specific cancer types and surgical procedures, such as colon and colorectal cancer, gastric adenocarcinoma, esophageal cancer, and gastrointestinal surgery. Including more specific recommendations and findings for different cancer types and surgical interventions can enhance the practical applicability of the review article.

Reply 2: We thank the reviewer for the valuable feedback provided. We appreciate the insightful suggestion to include more specific recommendations and findings related to immunonutrition and microbiota modulation in various cancer types and surgical procedures. In response, we would like to clarify that we did indeed touch upon the impact of immunonutrition and microbiota modulation in the other cancer types and surgical procedures in the article, such as hepatectomy and pancreatoduodenectomy for hepatocellular and pancreatic cancer. However, the available data limited us from providing a more detailed description and specific recommendations for the use of probiotics in surgical interventions. In this context, further research and robust clinical trials to refine recommendations are needed.

Reviewer 3 Report

Massive correction is required to improve the written English. Most of time the sentences are puzzling, and it makes the manuscript less informative and not organized. 

The review is rather short and does not provide useful insights on the use of immunonutrition in GI care. Most of the given information and points discussed are superficial, more in-depth discussion and insights given by the authors are expected, but it is not seen throughout the review

All the diagrams provide too simple information. 

Section 6 should be removed.

Massive correction is required. Most of the sentences are confusing. eg. line 31-34. Long sentence and the meaning is not clear. Line 34-35 - bad grammar. Repeated mistakes are seen throughout the manuscript. Major paraphrasing and sentence restructuring are required

Author Response

Point 1: The review is rather short and does not provide useful insights on the use of immunonutrition in GI care. Most of the given information and points discussed are superficial, more in-depth discussion and insights given by the authors are expected, but it is not seen throughout the review.

Reply 1: We sincerely appreciate the reviewer's feedback and insightful comments regarding the depth and coverage of our review article on immunonutrition in GI care. Accordingly, we have made substantial improvements to enhance the comprehensiveness and depth of the paper. Firstly, we have included a comprehensive section (3.2) on the Calorie Restriction and fasting as a potential nutritional strategy for GI cancer patient. Additionally, we have delved into a deeper exploration of the microbiota's crucial role in modulating the immune response, shedding more light on this intricate interplay and its implications for GI cancer patients (section 4 “Immunonutrition and Gut Microbiota Modulation”, page 8, lines 254-282 and page 12, lines 497-532). Furthermore, we have presented a more thorough examination of the safety and efficacy of biotics in specific GI cancers and related surgical procedures, aiming to provide a more comprehensive understanding of their potential applications (subsection 4.2, lines 608-627). We hope these enhancements might have significantly bolstered the depth and value of the review, allowing for a more comprehensive analysis of immunonutrition in the context of gastrointestinal care.

Point 2: All the diagrams provide too simple information. 

Reply 2: We regret that the diagrams did not meet the reviewer’s expectations. Our intention with these artworks was to create simple, self-explanatory schemes that effectively convey the information. However, we understand the need for clarity and depth, and we are committed to addressing the reviewer's suggestion by making significant improvements to Figure 1.

Point 3: Section 6 should be removed.

Reply 3: Accordingly, we have removed this section.

Point 4: Comments on the Quality of English Language: Massive correction is required. Most of the sentences are confusing. eg. line 31-34. Long sentence and the meaning is not clear. Line 34-35 - bad grammar. Repeated mistakes are seen throughout the manuscript. Major paraphrasing and sentence restructuring are required.

Reply 4: We thank the reviewer for pointing out this observation. In response to this feedback, we have extensively worked on enhancing the overall language quality, focusing on sentence restructuring, paraphrasing, and grammar refinement throughout the manuscript. We hope these efforts have significantly elevated the clarity and coherence of the text.

Round 2

Reviewer 1 Report

The authors have already significantly improved the manuscript based on my comments.

One more minor comment for Figure 2: 

It seems like the text "1-4" is separate from the images above, and wholeness is lost, authors should improve the layout for this figure.

Author Response

We appreciate your valuable input, and we are pleased to inform you that we have addressed the issue to enhance the overall cohesiveness of the figure

Reviewer 3 Report

The authors have made significant improvement in the revised manuscript. In the abstract, the authors mentioned that the interaction between nutrients and microbiota can help to maintain individual's immune system, however, I feel this was not properly explained and discussed in the manuscript. Can you further clarify by giving specific examples in Section 4.2?

Author Response

Thank you for your insightful comment. We have added two paragraphs at the beginning of Section 4.1 and at the end of Section 4.2 to discuss the interaction between immunonutrients and microbiota, providing specific examples that illustrate how these interactions can contribute to maintaining an individual's immune system